# MSEARCHER: SELF-REFLECTIVE SEARCH AGENT EMPOWERED BY MONTE CARLO TREE SEARCH BASED DATA SYNTHESIS

## ABSTRACT

Recent advances in reinforcement learning (RL) have enabled large language models (LLMs) to perform multi-turn chain-of-thought (CoT) reasoning with tool use, where web search serves as the most critical tool for answering complex questions. However, most existing methods apply RL directly to off-the-shelf models without a supervised fine-tuning (SFT) cold start, resulting in unstable training and limited tool invocations. This difficulty is exacerbated by the high cost of curating long reasoning trajectories, which are expensive to annotate and prone to factual drift. We propose MSEARCHER, a two-stage trained search agent that combines reflective thinking with robust tool use for complex reasoning. A central contribution is an efficient data construction framework based on Monte Carlo Tree Search (MCTS), which produces self-reflective reasoning trajectories for the SFT cold start. This framework leverages both correct and flawed rollouts to generate natural and diverse reasoning data. We adopt a two-stage pipeline, first applying SFT with our constructed data and then further training the model with RL, achieving substantial improvements on multi-hop question answering: $67.6\%$ on HotpotQA and $52.0\%$ on Frames. These results highlight the importance of high-quality SFT in stabilizing RL and equipping LLMs with robust long-horizon reasoning capabilities.

## 1 INTRODUCTION

Large language models (LLMs) have demonstrated remarkable progress across a wide spectrum of reasoning tasks (Hendrycks et al., 2021; Rein et al., 2023). , but they often suffer from hallucination—confidently producing factually incorrect answers due to outdated, incomplete, or erroneous parametric knowledge (Huang et al., 2025; Sardana, 2025). To mitigate this issue, Retrieval-Augmented Generation (RAG, Lewis et al., 2020) has become a standard practice, grounding model outputs in external evidence and substantially improving factual reliability (Gao et al., 2023; Cao et al., 2023; Xin et al., 2024).

Recent studies have begun to regard retrieval as an external tool (Li et al., 2025b), which can be deliberately triggered by the model whenever needed. This perspective extends the RAG paradigm into the broader framework of tool-augmented reasoning, where LLMs are trained not only to consume retrieved evidence but also to decide when and how to call external resources. To strengthen this ability, reinforcement learning (RL) has been introduced as a key training paradigm (Jin et al., 2025; Song et al., 2025; Zheng et al., 2025; Gao et al., 2025). By optimizing against task-specific rewards, RL enables models to improve their decision-making across multi-turn tool-use trajectories.

However, previous works often suffer from several drawbacks: (1) insufficient tool invocation, where the reasoning agent tends to halt after only one or two search calls, leading to inadequate context for problem solving; and (2) unstable training dynamics, as models without fine-tuning frequently produce outputs with invalid formats, leading to collapse in the early stages (Jin et al., 2025).

Inspired by recent advances in post-training (Chu et al., 2025; Ye et al., 2025), we posit that a moderate fine-tuning phase can serve as an effective cold start, equipping the model with fundamental tool-use capabilities and a stable reasoning scaffold. In this paper, we propose MSEARCHER, a two-stage trained search agent that combines reflective thinking with robust tool use for complex

reasoning, highlighted by an efficient data construction framework based on Monte Carlo Tree Search (MCTS) to produces self-reflective reasoning trajectories for SFT cold start. In contrast to prior works (Lee et al., 2025; Li et al., 2025a) that depends on expert large reasoning models (LRMs) to directly produce full trajectories, our framework decomposes the question into smaller sub-problems and solves them step by step. Our key intuition is to take advantage of the structure of MCTS. Rather than discarding low-quality outputs through rejection sampling, the data constructor leverages both correct and incorrect rollouts as complementary training signals. By contrasting flawed reasoning paths with more accurate trajectories, the framework induces natural self-reflection behavior and strengthens long-horizon reasoning. This property makes the data construction framework not only resource-efficient but also robust to noise.

To be more specific, the MCTS-based framework takes a multi-hop task as input, and we design the search tree at the granularity of task decomposition plans. Each node represents a complete plan containing multiple sub-tasks, and expansion corresponds to refining the plan by further decomposing one currently divisible sub-task. This unique design reduces the required search depth, while still enabling exponential growth in the breadth of candidate decompositions. Additionally, when the simulation ends, we identify and synthesize multiple forms of self-reflection across different branches and rollouts, focusing on retrieval, planning, and reasoning errors. As a result, our framework explores the reasoning space more efficiently without sacrificing coverage.

In our experiments, we adopt a two-stage training strategy. We first perform SFT on the our constructed data, and then apply RL to further enhance performance in real-world environments. Extensive results show that this two-stage training significantly outperforms previous one-stage methods, demonstrating the effectiveness of our constructed data. In summary, our contributions are three-fold: (1) we propose a novel data construction framework that generates self-reflective long CoT reasoning data; (2) we introduce a two-stage tool-augmented reasoning agent for complex tasks, equipped with a real-world search environment; and (3) we demonstrate the effectiveness of cold-start SFT through extensive experiments, achieving $67.6\%$ on HotpotQA and $52.0\%$ on Frames, while observing explicit self-verification behaviors.

## 2 PRELIMINARY

In this section, we first outline the task definition and then present the basic idea of Monte Carlo Tree Search that serves as the foundation of our framework.

### 2.1 TASK DEFINITION

In this work, we focus on the open-domain multi-hop question answering task, which requires performing multiple reasoning steps across different documents. A multi-hop question $q$ can be decomposed into a sequence of sub-questions, $q = \{q_1, q_2, \cdots, q_n\}$, arranged in topological order. In the first stage of SFT, our goal is to construct valid reasoning chains augmented with tool invocations for a given question as training data. Specifically, we adopt ReAct (Yao et al., 2023) as the underlying reasoning framework. At iteration $i$ of the reasoning chain, the LLM generates a thought $\tau_i$, executes an action (e.g., a tool call) $a_i$, and then receives an environmental observation $o_i$. The iteration ends when the LLM outputs "final answer" as the action. A complete trajectory $\mathcal{R}_T$ with $T$ iterations can be represented as:

$$\mathcal{R}_T = (\tau_0, a_0, o_0, \cdots, \tau_i, a_i, o_i, \cdots, \tau_T, a_T) \tag{1}$$

In the RL stage, the thought $\tau_i$ and action $a_i$ are sampled from the policy model based on previous context, i.e., $\pi(\tau_i, a_i | \mathcal{R}_{i-1})$.

### 2.2 MONTE CARLO TREE SEARCH(MCTS)

A typical MCTS algorithm builds a search tree $\mathcal{T}$ based on a policy model $\pi_\theta$, which is usually the reasoning backbone LLM for data construction. Each node in $\mathcal{T}$ is $n_t = [s_t, a_t, N(n_t), V(n_t)]$, where $s_t$ stands for the current state, $a_t$ is the possible next action, $N$ is the number of visits, and $V$ is the value function - the accuracy. In our case, the state $s_t$ corresponds to the current complete plan consisting of multiple sub-questions, while the action $a_t$ represents performing a single decomposition step to further refine this plan. For the $t$-th simulation, our MCTS runs the four standard operations:

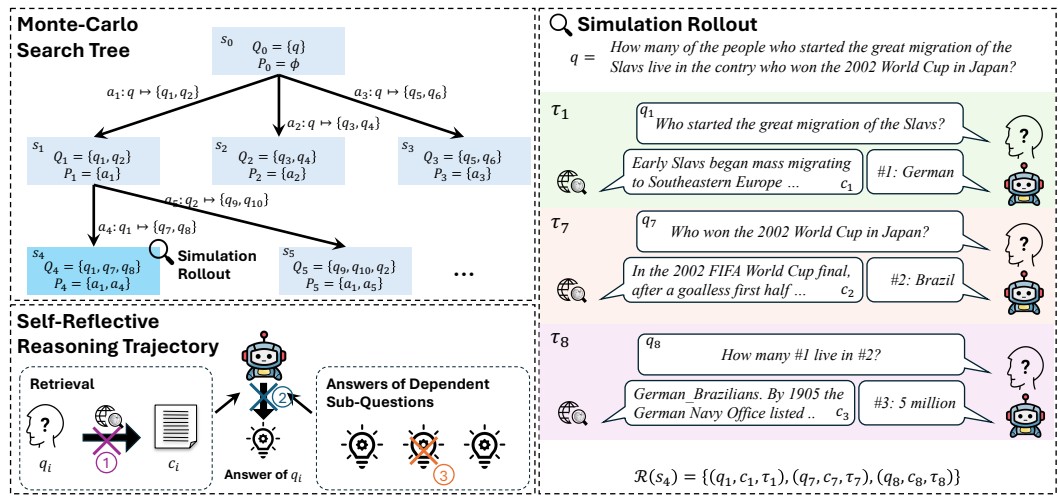

Figure 1: Illustration of MSearcher's data construction framework. Crosses and numbers denote error types, corresponding to retrieval error, reasoning error and decomposition error, respectively.

(1) **Selection** means to select a node with the highest value in the tree. In this paper, we adopt the UCT(Upper Confidence bounds applied to Trees) score (Kocsis & Szepesvári, 2006) as follows:

$$UCT(n_t) = V(n_t) + \omega\sqrt{\frac{2\log N(p)}{N(n_t)}} \tag{2}$$

where $\omega$ is the factor that controls exploration and exploitation and $p$ is the parent node of $s_t$.

(2) **Expansion** involves generating new states by taking plausible actions. In our setting, the model performs a decomposition on one of the sub-questions within $s_t$ to produce refined plans.

(3) **Simulation** means to perform a rollout for the current decomposition plan by querying a rollout model, with the help of the tools needed to get an answer and compute the reward $r$.

(4) **Backpropagation** collect the simulation reward $r$ and update the value $V(s)$ for all nodes along the path from the current node to the root.

$$N_{new}(s) = N_{old}(s) + 1,$$
$$V_{new}(s) = \frac{V_{old}(s)N_{old}(s) + r}{N_{new}(s)} \tag{3}$$

## 3 MSEARCHER

In this section, we introduce the main components of our data construction framework and training method. To be specific, in section 3.1, we introduce the detailed definition of our monte carlo search tree, how to do rollout and simulation, and synthesize reasoning trajectories from the search tree, as demonstrated in Figure 1; in section 3.2, we give details of building the reasoning agent.

### 3.1 SELF-REFLECTIVE DATA CONSTRUCTION

#### 3.1.1 NODE DEFINITION AND EXPANSION

In our search tree, each node represents a partial decomposition of a multi-hop question into sub-questions. Let a multi-hop question be $q$, and let the set of its current sub-questions at step $t$ be $Q_t = \{q_1, q_2, \ldots, q_K\}$. Then, a node $s_t$ in the search tree is defined as:

$$s_t = (Q_t, P_t), \tag{4}$$

where $Q_t$ is the current set of sub-questions and $P_t$ is the decomposition history that led to this state.

An action $a_t$ at node $s_t$ corresponds to performing a decomposition on *one* sub-question $q_i \in Q_t$, splitting it into exactly two smaller sub-questions:

$$a_t : q_i \mapsto \{q_i^1, q_i^2\}. \tag{5}$$

The resulting next state $s_{t+1}$ is:

$$s_{t+1} = (Q_t \setminus \{q_i\}) \cup \{q_i^1, q_i^2\}, \quad P_{t+1} = P_t \cup \{a_t\}. \tag{6}$$

A node is considered as leaf if no sub-question in $Q_t$ can be further decomposed, *i.e.*, all remaining sub-questions are atomic, judging by the policy model. By restricting each decomposition to split a sub-question into exactly two smaller tasks, we keep the expansion task simple enough for the policy model to handle, avoiding reliance on a powerful LRM to do this job.

In practice, we define an expansion width coefficient $w$ and distribute it evenly across the decomposable sub-questions $q_i$ in $Q_t$, such that $w = \sum_i w_i$. We then prompt the policy model to generate $w_i$ distinct decompositions for each $q_i$ in a single response, preventing the model from producing identical decomposition plans across different branches. This design means that in our framework we actually perform multiple expansions and rollouts within a single simulation round.

### 3.1.2 TREE POLICY

Following the standard MCTS framework, our tree policy iteratively selects the most promising child node at each level. Formally, given a current node $s_t$, we select the child node $s_{t+1}$ that maximizes the selection value:

$$s_{t+1} = \arg \max_{s' \in \text{Children}(s_t)} \text{UCT}(s') \tag{7}$$

where $\text{UCT}(s')$ is computed as in Eq. 2, where the coefficient $\omega$ is set to 0.6 in practice. This selection continues recursively until a leaf node (a node without children) is reached.

If the leaf node is expandable according to our decomposition rules, it undergoes expansion; otherwise, we skip expansion and proceed directly to the simulation phase. We keep a record of each sub-questions in $Q_t$ of node $s_t$ on whether it is decomposable by prompting the policy model. This ensures that only nodes with potential for further task decomposition are explored, while fully expanded or terminal nodes are directly evaluated in rollouts.

### 3.1.3 ROLLOUT

In the simulation rollout step, the model executes reasoning for a given node in the search tree to produce the results of its sub-questions. Specifically, for the current node state $s_t$, we maintain its corresponding set of sub-questions $Q_t = \{q_1, q_2, \ldots, q_K\}$. Each sub-question may depend on the results of other sub-questions, indicated by references such as #i, denoting a dependency on the $i$-th sub-question. These dependencies form a directed acyclic graph (DAG) over $Q_t$, which is traversed using topological sorting to obtain a valid execution order $\{q_1, q_2, \ldots, q_K\}$ that satisfies all dependencies.

For each sub-question $q_j$ in this order, the rollout proceeds in two steps. First, a retrieval operation is performed to collect relevant context or evidence from external sources, such as a web search. Let $c_j = \text{Retrieve}(q_j)$ denote the retrieved context. Second, the rollout model generates a response $\tau_j$ conditioned on both the retrieved context $c_j$ and the answers of any dependent sub-questions:

$$\tau_j = \text{RolloutModel}\Big(q_j \mid c_j, \{\tau_i \mid q_i \in \text{Dep}(q_j)\}\Big), \tag{8}$$

where $\text{Dep}(q_j)$ denotes the set of sub-questions that $q_j$ depends on. Iterating over all sub-questions produces the complete rollout trajectory for the node $s_t$:

$$\mathcal{R}(s_t) = \{(q_1, c_1, \tau_1), (q_2, c_2, \tau_2), \cdots, (q_K, c_K, \tau_K)\}. \tag{9}$$

This iterative approach ensures that each sub-question is answered in a contextually grounded manner while respecting the dependencies among sub-questions. By combining retrieval and reasoning at each step, the model produces a coherent, multi-step solution for the node.

### 3.1.4 Self-Reflective Reasoning Trajectory

When the simulation ends, we collect all rollout trajectories from the leaf nodes of the search tree and analyze them to construct self-reflective reasoning data.

First, we identify trajectories that yield correct final answers, marking them as *correct trajectories*, which can be directly textualized into supervised training data and used for comparison with other trajectories. For trajectories that yield incorrect final answers, we perform structured error detection to extract self-reflection signals. Specifically, we categorize errors into three categories. When we detect these error, we pair incorrect trajectories with correct ones and ensemble them into *self-reflective trajectories*. The three categories are as follows:

**Retrieval Error.** This error arises when two rollouts correspond to the same leaf node (i.e., identical decomposition and sub-question set). Formally, let $\mathcal{R}^+(s_t)$ denote a correct rollout trajectory and $\mathcal{R}^-(s_t)$ a wrong one. Suppose they are aligned in the first $j - 1$ step and the difference first occurs at step $j$, where the retrieved context $c_j$ of $\mathcal{R}^+(s_t)$ differs from that of $\mathcal{R}^-(s_t)$, leading to downstream reasoning failure. In this case, we construct the reflective trajectory $\mathcal{R}^{\mathrm{ref}}(s_t)$ as follows:

$$\mathcal{R}^{\mathrm{ref}}(s_t) = \mathcal{R}^-(s_t)[: j] \, \| \, (q_j, c_j^+, \tau_j^+) \, \| \, \mathcal{R}^+(s_t)[j + 1 :],$$

where $c_j^+$ and $\tau_j^+$ are the corresponding context and answer in the correct trajectory $\mathcal{R}^+$.

**Reasoning Error.** This occurs when two rollouts of the same leaf node share the same decomposition and retrieval contexts up to step $j$, but the rollout model produces different answers for $\tau_j$. Such divergence indicates a reasoning error. Formally, let $\mathcal{R}^+(s_t)$ and $\mathcal{R}^-(s_t)$ denote the correct and wrong trajectory, respectively, then we define the reflective trajectory as:

$$\mathcal{R}^{\mathrm{ref}}(s_t) = \mathcal{R}^-(s_t)[: j] \, \| \, (q_j, c_j, \tau_j^+) \, \| \, \mathcal{R}^+(s_t)[j + 1 :],$$

where $\tau_j^+$ denotes the reasoning outcome for step $j$ of the correct trajectory $\mathcal{R}^+$.

**Decomposition Error.** This error type arises across different branches of the search tree, where one branch produces correct rollouts while another fails. Since both branches have undergone multiple rollouts, we attribute the discrepancy to different task decompositions. Let $\mathcal{R}^+(s_m)$ and $\mathcal{R}^-(s_n)$ denote the correct and wrong rollout trajectory, respectively. Suppose that $s_m$ and $s_n$ share the same decomposition plan up to step $j$, then we can define the reflective trajectory as:

$$\mathcal{R}^{\mathrm{ref}}(s_t) = \mathcal{R}^-(s_t)[: j] \, \| \, \mathcal{R}^+(s_t)[j :],$$

where $j$ is the first step after which the decompositions differ.

In this way, the reflective trajectory explicitly encodes both the erroneous reasoning context from $\mathcal{R}^-$ and the corrective signal from $\mathcal{R}^+$. Finally, all trajectories—both correct and reflective—are textualized into natural language supervision. We achieve this by joining the steps with a diverse set of pre-defined and paraphrased connective sentences, thereby generating coherent reasoning chains that capture explicit self-reflection and error correction. The detailed synthesized reasoning trajectories can be found in Appendix A.

## 3.2 Agent Training

In this section, we present our two-stage training pipeline for building a tool-augmented reasoning agent, consisting of (1) a cold-start supervised fine-tuning stage on the constructed data, followed by (2) reinforcement learning to align the agent with real-world search environments.

### 3.2.1 Cold-Start Supervised Fine-Tuning

In the cold-start supervised fine-tuning (SFT) stage, we initialize the policy model with a reliable reasoning and retrieval strategy derived from our reflective trajectories. This step equips the model with stable tool-use patterns and self-correction signals, which are crucial for mitigating unstable exploration and avoiding collapse in the early stages of RL training.

Consistent with recent empirical studies (Zhang et al., 2025), we exclude the loss contributions from these external observations . Given an input reasoning trajectory $\mathcal{R}_T$ as in Eq. 1, the model

parameters $\theta$ are optimized by a masked auto-regressive negative log-likelihood:

$$\mathcal{L} = -\frac{\sum_{t=1}^{T} \mathbb{I}[\tau_t, a_t \notin \mathcal{O}] \cdot \log p_\theta(\tau_t, a_t | \tau, a, o_{<t})}{\mathbb{I}[\tau_t, a_t \notin \mathcal{O}]} \tag{10}$$

where $\mathbb{I}[\tau_t, a_t \notin \mathcal{O}]$ is an indicator function that masks the loss of content in the environment observations $\mathcal{O} = \{o_i | i = 0, \cdots, T\}$. This masking ensures that the model is optimized only on the supervised signals generated by the agent itself, without being penalized for content from external tools. In practice, we also split the dialogue into smaller segments, allowing the model to learn reasoning step by step rather than attempting to solve the entire problem in a single generation.

### 3.2.2 Reinforcement Learning with DAPO

After the cold-start stage, we further enhance the model's capabilities using reinforcement learning (RL). We adopt the Dynamic sAmpling Policy Optimization (DAPO) algorithm (Yu et al., 2025), which builds upon Group Relative Policy Optimization (GRPO) (Shao et al., 2024) by incorporating a few modifications: clip-higher, removal of the KL loss, switching to token-level loss, dynamic sampling, and reward shaping. Considering training efficiency and effectiveness, we apply all modifications except dynamic sampling. Given a batch of examples $(q, a) \sim \mathcal{D}$ and a group of sampled responses $\{o_i\}_{i=1}^{G}$ from the old policy $\pi_{old}$, the optimizing object of DAPO is:

$$\mathcal{J}_{DAPO} = \mathbb{E}_{(q,a)\sim\mathcal{D}, \{o_i\}_{i=1}^{G} \sim \pi_{old}(\cdot|q)}$$

$$[\frac{1}{\sum_{i=1}^{G} |o_i|} \sum_{i=1}^{G} \sum_{t=1}^{|o_i|} \min(r_{i,t}(\theta)\hat{A}_{i,t}, clip(r_{i,t}(\theta), 1 - \epsilon_{low}, 1 + \epsilon_{high})\hat{A}_{i,t})] \tag{11}$$

where the $r_{i,t}(\theta)$ is the importance sampling ratio between the new policy and the old policy at time $t$, and $\hat{A}_{i,t}$ denotes the estimated advantage. The parameter $\epsilon_{high}$ and $\epsilon_{low}$ controls the upper and lower clip bound, and in practice they are set to 0.28 and 0.2, respectively.

### 3.2.3 Reward Design

In this work, we consider the sum of three major parts as our reward. First, the primary reward is the answer correctness $R_{correct}$, evaluated by F1-score. The Second, following by DAPO, is the length penalty $R_{length}$. And finally, we also considered the format penalty $R_{format}$:

$$R_{format} = \min(|-1 - (R_{correct} + R_{length})|, -0.2) \tag{12}$$

## 4 Experiment

### 4.1 Dataset

In this work, a lot of different datasets and benchmarks are used for different purpose. First, in the SFT data construction, we use Musique (Trivedi et al., 2022) as the seed dataset, mainly because it offers oracle question decomposition, which can serve as golden reference when policy model fails. It also provides a limited context to retrieve from, which greatly reduced the computational cost.

With respect to the RL, we adopt the same training corpus with DeepResearcher (Zheng et al., 2025), which includes NaturalQuestion (NQ) (Kwiatkowski et al., 2019), TriviaQA (TQ) (Joshi et al., 2017), HotpotQA (Yang et al., 2018) and 2WikiMultiHopQA (2Wiki) (Ho et al., 2020).

Finally, for the evaluation, we also follow DeepResearcher to evaluate the model in both in-domain (ID) and out-of-domain (OOD) settings. We include NQ, TQ, HotpotQA, and 2Wiki for the ID setting, and Bamboogle (Press et al., 2023) and PopQA (Mallen et al., 2022) for the OOD evaluation. Furthermore, we also include FanoutQA (Zhu et al., 2024), Frames (Krishna et al., 2025) and GAIA (Mialon et al., 2024) as three more challenging benchmarks. For GAIA, we use the 103 text-only examples as validation set (Gao et al., 2025).

### 4.2 Baselines

In this work, we adopt two categories of baselines with different paradigms. For the **Prompt-Based** baselines, we have: (1) **CoT**, where the model employs basic Chain-of-Thought (Wei et al., 2022)

reasoning; (2) **RAG**, which combines the CoT with retrieved context from local resources. (3) **Search-o1** (Li et al., 2025b) is a prompt-based multi-step reasoning agent that can generate search queries and retrieve external information to assistant question answering. We implement this baseline with the web search as tested in DeepResearcher.

For the **Training-Based** baselines, we includes: (4) **Search-r1** (Jin et al., 2025) introduces a reinforcement learning approach for question answering, where both training and inference rely on a local retriever to search for relevant information. (5) **R1-Searcher** (Song et al., 2025) proposes a similar approach with Search-r1, but differs in search strategy. It search the wikipedia via Bing and summaries the top three pages. (6) **DeepResearcher** (Zheng et al., 2025) proposes to improve the agent's search ability by training with a real-world search engine and browser. (7) **ASearcher** (Gao et al., 2025) introduce a search agent trained by large-scale RL with autonomously synthesized high-quality and challenging QAs.

### 4.3 Metrics

We adopt two complementary of metrics in this work:

**Rule-based Metrics** Given the scale and computational cost of training, we primarily adopt the rule-based F1 score for reward computation and evaluation during RL training. Both the predicted answer and the ground truth are normalized—by removing punctuation and standardizing letter case—before calculating the F1 score.

**Model-based Evaluation** Rule-based metric is not accurate in some cases such as long answer. Therefore, we adopt a LLM-as-Judge score by prompting a strong LLM to ask whether the answer is correct. Specifically, we follow DeepResearcher, leveraging GPT-4o-mini as the judge model.

### 4.4 Implementation

For all the models in SFT and RL training, we use Qwen2.5-7B-Instruct or Qwen2.5-14B-Instruct (Yang et al., 2024) as the training backbones. The SFT training is completed using the OpenRLHF (Hu et al., 2024) framework and the RL training is conducted with the Slime (Zhu et al., 2025) framework. In the SFT, we use a total batch size of 32 and train for 5 epochs with learning rate of 5e-6, warmup ratio of 0.05, and a sequence length of 16k. In the RL, we sample 128 examples in rollout step, sample 8 responses for each prompt, and then update the policy model with a mini-batch size of 256. Each rollout response is limited to up to 10 tool calls and a total length of 16k.

For the data construction framework, we use QwQ-32B (Team, 2025) as both the policy model and rollout model. Since the the dataset Musique provides contexts, we embed the context with Zhipu Embedding-3 (big, 2025b) and use faiss algorithm (Douze et al., 2024) for retrieval.

With respect to the search tool, for the local RAG in the data construction and for some baseline such Search-r1, we adopt the retrieval strategy as stated above; for the online web search, we adopt the Zhipu search tool (big, 2025a) as the primary search engine interface for searching and page reading.

## 5 Result

### 5.1 Main Results

Table 1 demonstrates the performance of MSEARCHER and other baselines on both ID and OOD benchmarks. It can be seen from the result that, MSEARCHER outperforms other baselines on most benchmarks, and achieves the best average results in both ID and OOD settings. This suggest that, MSEARCHER not only not only learns to leverage search tools effectively within the training domain, but also generalizes well to unseen data. Specifically, compared with DeepResearcher, which shares the same training data and web environment, our method achieves an average improvement of 4.2% in the ID setting and 1.4% in the OOD setting, highlighting the benefits of our self-reflective reasoning data and SFT cold start. With respect to ASearcher, it outperforms ours on multi-hop datasets such as HotpotQA and 2Wiki. However, its performance drops significantly on NQ, likely because their model is trained on self-curated complex reasoning data, constructed primarily from

| Method | Inference Environment | In Domain | | | | | Out of Domain | | |
|---|---|---|---|---|---|---|---|---|---|
| | | NQ | TQ | Hotpot | 2Wiki | Avg | Bamb | PopQA | Avg |
| CoT | - | 32.0 | 48.2 | 27.9 | 27.3 | 33.9 | 21.6 | 15.0 | 18.3 |
| CoT+RAG | Local RAG | 59.6 | 75.8 | 43.8 | 24.8 | 51.0 | 27.2 | 48.8 | 38.0 |
| Search-o1 | Web Search | 55.1 | 69.5 | 42.4 | 37.7 | 51.2 | 53.6 | 43.4 | 48.5 |
| *7B Models* | | | | | | | | | |
| Search-r1 | Local RAG | 49.6 | 49.2 | 52.5 | 48.8 | 50.0 | 47.2 | 44.5 | 45.9 |
| R1-Searcher | Web Search | 52.3 | 79.1 | 53.1 | 65.8 | 62.6 | 65.6 | 43.4 | 54.5 |
| DeepResearcher | Web Search | 61.9 | 85.0 | 64.3 | 66.6 | 69.5 | 72.8 | 52.7 | 62.8 |
| ASearcher* | Web Search | 55.4 | 85.7 | 67.2 | 73.3 | 70.4 | 72.0 | 48.9 | 60.5 |
| *14B Models* | | | | | | | | | |
| Search-r1 | Local RAG | 62.9 | 82.6 | 65.8 | 56.4 | 66.9 | 64.3 | 53.8 | 59.1 |
| DeepResearcher | Web Search | 63.5 | 86.9 | 66.9 | 69.3 | 71.6 | 72.8 | 54.5 | 63.7 |
| ASearcher* | Web Search | 55.5 | 87.6 | **68.5** | **80.7** | 73.1 | **75.2** | 50.0 | 62.6 |
| MSearcher | Web Search | **68.7** | **87.9** | 67.6 | 77.4 | **75.3** | 74.5 | **56.8** | **65.6** |

Table 1: Main results on six multi-hop question answering benchmarks. Most baseline results are from Zheng et al. (2025) and Gao et al. (2025). The ASearcher is labelled with * because it is trained with its own curated RL data.

multi-hop sources (HotpotQA and 2Wiki), leading to poor generalization to other datasets such as NQ and PopQA.

Table 2 presents the performance of Deep-Research, Search-r1, and MSEARCHER, all using a 14B backbone and trained on the same data as in Table 1, evaluated on three challenging datasets that require extensive searches or more sophisticated reasoning. As shown, Search-r1 with its local retrieval system performs worse than the others. While DeepResearch and our method both achieve stable and competitive results, MSEARCHER consistently demonstrates a clear advantage.

| Model | FanoutQA | FRAMES | GAIA | Avg. |
|---|---|---|---|---|
| DeepResearcher-14B | 45.2 | 48.6 | 38.7 | 44.2 |
| Search-r1-14B | 13.3 | 22.4 | 11.8 | 15.8 |
| MSearcher | **47.6** | **52.0** | **40.5** | **46.8** |

Table 2: Evaluation on three more challenging benchmarks, MSEARCHER and DeepResearch use web search, while Search-r1 uses local retrieval system.

## 5.2 ABLATION STUDY

This section presents ablation results that examine how different components impact the overall performance of our framework.

**Effectiveness of Self-Reflective Signal**. In this experiment, we remove all examples containing self-reflective behaviors during the SFT stage, using only correct MCTS rollouts for data construction. As shown in Table 3, excluding self-correction data reduces the number of tool calls, since the average tool usage in the SFT training set becomes lower. Consequently, the final performance after RL training also declines.

| Method | HotpotQA | | Frames | |
|---|---|---|---|---|
| | Acc | Tool Call | Acc | Tool Call |
| MSearcher | 67.6 | 5.6 | 52.0 | 5.7 |
| *- w/o ref* | 65.7 | 4.7 | 50.4 | 4.5 |
| *- w/o sft* | 63.3 | 2.7 | 48.7 | 2.9 |

Table 3: Ablation results of MSearcher on SFT: without reflective data in SFT and without SFT.

**Model Performance without RL**. In this experiment, we evaluate the model's performance after SFT but before RL, compared to the vanilla instruction model. As shown in Table 4, the model's performance on HotpotQA drops by 4.86%, accompanied by an increase in the number of tool calls. This decline can be attributed to domain shift and a stronger bias toward tool usage introduced by the SFT data. Notably, this phenomenon disappears with the larger 14B backbone and in the in-domain evaluation on the Musique dataset.

## 5.3 TRAINING DYNAMICS

Figure 2 compares training rewards and tool calls of our method with and without SFT using qwen2.5-7B. Tool calls increase steadily in both cases, but SFT gives a higher initial tool usage, leading to better performance.

## 6 RELATED WORK

### 6.1 SEARCH AGENTS

Recent works have explored building agent workflows that allow large language models (LLMs) to interact with external tools for complex task solving. Representative prompt-based agent systems such as Search-o1 (Li et al., 2025b) and ReAgent (Zhao et al., 2025), while effective for rapid development, are constrained by the inherent capacity of the underlying LLM and are not easily improved through environment feedback. To address this, several studies have attempted to construct supervised fine-tuning (SFT) trajectories. For example, some works (Lee et al., 2025; Asai et al., 2024; Yu et al., 2024) employ large LLMs to synthesize retrieval and reasoning traces, which are then used to fine-tune smaller models. Most recently, reinforcement learning (RL) methods have been investigated to enhance LLM-based agents, and most prior works focus on multi-hop QA settings (Jin et al., 2025; Song et al., 2025; Zheng et al., 2025). Moreover, Tan et al. (2025) propose to use direct prompting to generate reasoning data and combines SFT with RL to improve search strategies.

| Model | HotpotQA | | Musique | |
|---|---|---|---|---|
| | Acc | Tool Call | Acc | Tool Call |
| **Qwen2.5-7B** | | | | |
| Instruct | 51.9 | 1.3 | 21.6 | 1.1 |
| SFT | 47.1 (-4.8) | 3.9 | 26.5 (+4.9) | 4.1 |
| **Qwen2.5-14B** | | | | |
| Instruct | 62.9 | 1.5 | 25.8 | 1.5 |
| SFT | 65.4 (+2.5) | 4.8 | 31.3 (+5.5) | 4.9 |

Table 4: Evaluation of vanilla Qwen2.5-7B/14B instruction models before and after SFT.

### 6.2 SYNTHETIC TRAJECTORIES FOR SEARCH AGENTS

Large-scale manual annotation is often costly and inflexible, motivating the use of synthetic data generation as a scalable alternative for training search agents. Previous works like ReaRAG (Lee et al., 2025) and RAG-R1 (Tan et al., 2025) adopt the prompt-based method to instruct the large reasoning model (LRM) to produce reasoning data in predefined format. Recent methods create realistic QA trajectories for SFT by simulating interactions with real-world web pages, with LRMs assisting in curating the collected data (Wu et al., 2025; Li et al., 2025a). However, these methods often rely on powerful LRMs and large number of generations for reject sampling.

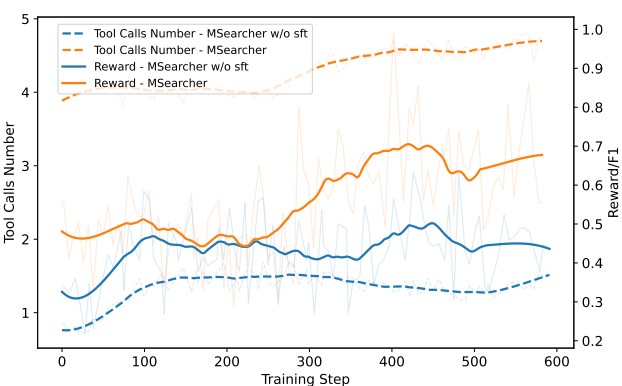

Figure 2: Training dynamics of Reward and Number of tool calls of MSearcher with/witout SFT.

## 7 CONCLUSION

In this work, we present MSEARCHER, a two-stage trained search agent that can perform robust and reflective multi-step reasoning. A core contribution is a MCTS–based framework that generates self-reflective reasoning trajectories for supervised fine-tuning, enabling stable RL training without large reasoning models. Experiments on multi-hop QA benchmarks show MSEARCHER outperforms strong baselines, demonstrating the value of high-quality SFT data and reflective reasoning.

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

## A PROMPTS AND DATA EXAMPLE

This section shows prompts used in data construction and model training.

---

**Data Construction: Question Decomposition**

Decompose the following question into TWO sub-questions, such that the original question can be broken down into two sequential sub-questions. You should use [E] to represent the answer entity of one sub-question that appears in another. For the two sub-questions you generate, one shouldn't contain [E] and the other one should contain [E]. By answering the one without [E] first and then the one with [E] (which will be replaced by the answer entity of the first sub-question), one should be able to get the answer to the original question. Some questions in the input might contain #1, #2, etc. to represent the answer of the previous question. You should ignore them and keep them as is in the sub-questions. Give exactly different decompositions. Follow the format below:
Input: If my future wife has the same first name as the 15th first lady of the United States' mother and her surname is the same as the second assassinated president's mother's maiden name, what is my future wife's name?
Output: Decomposition 1:
Q1: What's the first name of the 15th first lady of the United States' mother?
Q2: If my future wife's surname is the same as the second assassinated president's mother's maiden name, and her first name is [E], what is my future wife's name?
Decomposition 2:
Q1: Who's the second assassinated president of the United States?
Q2: If my future wife's surname is the same as [E]'s mother's maiden name, and her first name is the same as the 15th first lady of the United States' mother's first name, what is my future wife's name?
And so on...
Question to decompose:
Input: {}
Output:

---

**Data Construction: Rollout**

Answer the following quesiton with only a short and concise answer(entity). The given context might include some information about the answer, if not, use your own knowledge. The result should be exactly in the following format and nothing else, do not provide any other information, explanation or thinking:
Input: Question: Who is the president of the United States? Context: 1. The current 47th president of U.S. is Donald J. Trump.
Output: Donald Trump
Now answer the question:
Question: {}
Context: {}
Answer:

---

**Data Construction: Determine If Decomposable**

Given a natural language question, determine whether it is a atomic question that not be decomposed in a meaningful and straightforward way. Some questions in the input might contain #1, #2, etc. You should view them as some known and fixed entities.
Examples: 1. Input: How many Germans live in Paris in 2024?
Output: Yes. Rationale: The question cannot be meaningfully decomposed.
2. Input: When was Yimou Zhang's first movie as a director released?
Output: No. Rationale: The question can be meaningfully decomposed into two sub-questions; for example, "What is Yimou Zhang's first movie as a director?" and "When was it released?"
3. Input: Who founded the CS department of #1?
Output: Yes. Rationale: This question cannot be meaningfully decomposed. It's asking the name of a person that is fully specified.

You should just output "Yes" or "No" and nothing else. Don't give any explanation or reasoning.
Input: {}
Output:

---

Model Training: SFT Data Example

{"role": "system", "content": "You are a helpful assistant. Answer the following question step by step with the help of web search tools. You must conduct reasoning inside <think> and </think> first in every turn. After reasoning, if you find you lack some knowledge, you can call a search engine by <search> query </search> and it will return the top searched results between <tool_response> and </tool_response>. You can call the search function once in every turn. When you arrive at the final answer, output the final answer between <answer> and </answer> to finish the task. Do not rush to output the final answer so you could check it again midway. "}

{"role": "user", "content": "What followed the last person to live in versaille in the country that became allies with america after the battle of saratoga?Put the final answer within <answer></answer>. "}

{"role": "assistant", "content": "<think>To solve this, I will divide it into smaller questions: [1. Which country became allies with America? 2. What significant event led to #1 becoming allies with America? 3. What followed the last person to live in Versailles in the country that became allies with America after #2?]. As a first step, I need to answer the subquestion 1: Which country became allies with America?. Let us find supporting evidence with a search:</think> <search>Which country became allies with America?</search>"}

{"role": "tool", "content": "1. Southern Europe: The outbreak of World War I in 1914 was precipitated by the rise of nationalism in Southeastern Europe as the Great Powers took up sides. The Allies defeated the Central Powers in 1918. During the Paris Peace Conference the Big Four imposed their terms in a series of treaties, especially the Treaty of Versailles."}

{"role": "assistant", "content": "<think>Based on what I found and already know, the answer is United Kingdom. Next, to address the subquestion 2: What significant event led to #1 becoming allies with America?, based on previous sub-questions, which turns to: What significant event led to United Kingdom becoming allies with America?.</think> <search>What significant event led to United Kingdom becoming allies with America?</search>"}

{"role": "tool", "content": "1. France in the American Revolutionary War: The British had taken Philadelphia in 1777, but American victory at the Battle of Saratoga brought back hope to the Patriots and enthusiasm in France. The army of Burgoyne surrendered to American forces after Saratoga and France realized that the United States could be victorious. The king directed Vergennes to negotiate an alliance with the Americans."}

{"role": "assistant", "content": "<think>The information points to the answer being Battle of Saratoga. At the end, I'll think through this next part: 3: What followed the last person to live in Versailles in the country that became allies with America after #2?, based on previous sub-questions, which turns to: What followed the last person to live in Versailles in the country that became allies with America after Battle of Saratoga?. To continue, I will look for relevant content:</think><search>What followed the last person to live in Versailles in the country that became allies with America after Battle of Saratoga?</search>"}

{"role": "tool", "content": "Michael Manning (murderer): Michael Manning was an Irish murderer who became the twenty - ninth and last person to be executed in the Republic of Ireland."}

{"role": "assistant", "content": "<think>From the evidence gathered, it seems the answer is Michael Manning (murderer). Wait, maybe I need to reconsider this more carefully. The supporting content might not be useful here. I should revise my search approach. Rephrase the search query.</think><search>Who succeeded the final inhabitant of Versailles in the nation that allied with the United States following the Battle of Saratoga?</search>"}

{"role": "tool", "content": "1. Battles of Saratoga: Once news of Burgoyne's surrender reached France, King Louis XVI decided to enter into negotiations with the Americans that resulted in a formal Franco - American alliance and French entry into the war. This moved

the conflict onto a global stage. As a consequence, Britain was forced to divert resources used to fight the war in North America to theaters in the West Indies and Europe, and rely on what turned out to be the chimera of Loyalist support in its North American operations. Being defeated by the British in the French and Indian War more than a decade earlier, France found an opportunity of revenge by aiding the colonists throughout the Revolutionary War. Prior to the Battle of Saratoga, France did n't fully aid the colonists. However, after the Battles of Saratoga were conclusively won by the colonists, France realized that the Americans had the hope of winning the war, and began fully aiding the colonists by sending soldiers, donations, loans, military arms, and supplies."}
{"role": "assistant", "content": "<think>Based on the search results and my prior knowledge, now I can answer the question, and the answer in French Republic. </think> <answer>French Republic</answer>"}

---

**Model Training: RL Prompt**

System:
You are a helpful assistant. Answer the following question step by step with the help of web search tools. You must conduct reasoning inside <think> and </think> first in every turn. After reasoning, if you find you lack some knowledge, you can call a search engine by <search> query </search> and it will return the top searched results between <tool_response> and </tool_response>. You can call the search function once in every turn. When you arrive at the final answer, output the final answer between <answer> and </answer> to finish the task. Do not rush to output the final answer so you could check it again midway.

User:
Question: {Question}

## B  CASE STUDY

As shown in the case below, MSEARCHER behaves in an expected ReAct pattern. We can observe that the agent sometimes tends to over-decompose a question, such as the second sub-question: Who is #1, making the reasoning chain deviate from the correct path. But fortunately, we can observe that MSEARCHER is able to realize its mistakes in decomposition and run it again to get the correct answer.

---

**Case Study**

**System:**
You are a helpful assistant. Answer the following question step by step with the help of web search tools. You must conduct reasoning inside <think> and </think> first in every turn. After reasoning, if you find you lack some knowledge, you can call a search engine by <search> query </search> and it will return the top searched results between <tool_response> and </tool_response>. You can call the search function once in every turn. When you arrive at the final answer, output the final answer between <answer> and </answer> to finish the task. Do not rush to output the final answer so you could check it again midway.

**User:**
What Japanese unit was commanded by the admiral targeted by Operation Vengeance?

**Assistant:**
<think> To solve this, I will divide it into smaller questions:
1. Who was the target of Operation Vengeance?
2. Who is #1?

3. What Japanese unit was commanded by the admiral who targeted #2?
Let us start by, I will address subquestion 1, which is: Who was the target of Operation Vengeance?.
</think>
<search> Who was the target of Operation Vengeance? </search>

**Tool Response (Search and Read):**
<tool_response>
1. Operation Vengeance was the American military operation to kill Admiral Isoroku Yamamoto of the Imperial Japanese Navy on 18 April 1943
2. The man perhaps most associated with the "Day of Infamy," Pearl Harbor attack mastermind Admiral Isoroku Yamamoto, was killed fighting a war.
3. Capt. Thomas G. Lanphier Jr. claimed to have shot down Yamamotoś plane, killing him in the process, but the evidence indicates it was not Lanphier.
...
</tool_response>

**Assistant:**
<think> Based on the search results and my prior knowledge, the answer is Admiral Isoroku Yamamoto.
Moving on, To begin with, I need to answer the subquestion 2: Who is #1?, based on previous sub-questions, which turns to: Who is Admiral Isoroku Yamamoto?.
</think>
<search> Who is Admiral Isoroku Yamamoto? </search>

**Tool Response (Search and Read):**
<tool_response>
1. Isoroku Yamamoto was a Marshal Admiral of the Imperial Japanese Navy (IJN) and the commander-in-chief of the Japanese Combined Fleet during World War II.
2. Yamamoto Isoroku was a Japanese naval officer who conceived of the surprise attack on the U.S. naval base at Pearl Harbor on Dec. 7, 1941.
3. Isoroku Yamamoto was a brilliant mind and a respected leader, compelled by loyalty to his country to enter a war he knew they could not win.
4. This essay will explore Yamamotoś military career in the context of Imperial Japanś aggressive expansion into Asia beginning in the 1890s.
...
</tool_response>

**Assistant:**
<think> From the evidence gathered, it seems the answer is Isoroku Yamamoto.
Hmm, this might need a second look before I proceed. I may have broken down the question incorrectly—time to rethink the subquestions.
Plan again:
1. Who was the target of Operation Vengeance? 2. What Japanese unit was commanded by the admiral #2 who was targeted?.
The first question is answer in previous turns, I will start by address subquestion 2, which is: What Japanese unit was commanded by the admiral Yamamoto Isoroku who was targeted?.
To continue, I will look for relevant content:
</think>
<search> What Japanese unit was commanded by the admiral Yamamoto Isoroku who was targeted? </search>

**Tool Response (Search and Read):**
<tool_response>
1. Isoroku Yamamoto was a Marshal Admiral of the Imperial Japanese Navy (IJN) and the commander-in-chief of the Japanese Combined Fleet during World War II.
2. Operation Vengeance was the American military operation to kill Admiral Isoroku Yamamoto of the Imperial Japanese Navy on 18 April 1943.

3. Commanding the Japanese Combined Fleet was Admiral Isoroku Yamamoto who viewed Midway as a way to draw out and destroy the US Pacific Fleetś aircraft carrier.
...
</tool_response>

---

**Assistant:**
<think> Based on the search results and my prior knowledge, now I can answer the question, and the answer in Japanese Combined Fleet.
</think>
<answer> Japanese Combined Fleet </answer>

