# OpenReview forum: "MSearcher: Self-Reflective Search Agent Empowered by Monte Carlo Tree Search Based Data Synthesis"
_ICLR.cc/2026/Conference — ICLR 2026 Conference Withdrawn Submission_

### Official Review · Reviewer_xeVA · 2025-10-27

**Soundness:** 3
**Presentation:** 3
**Contribution:** 3
**Rating:** 6
**Confidence:** 3

**Summary:**

This paper proposes MSearcher, which is a self-reflective search agent for open-domain multi-hop question answering task. The agent is trained with Supervised FineTuning (SFT) first, and then trained with Reinforcment Learning (RL) using Dynamic sAmpling Policy Optimization (DAPO). The data used for training is synthesized by rollout using Monte Carlo Tree Search (MCTS), where each node is a partition of tasks and the leaves are lists of atomic tasks. The atomic tasks are then answered by the rollout model in a logically topological order to form a trajectory towards the final answer. The trajectories that leads to the correct final answer are used for supervised learning and comparison with other trajectories; the rest are categorized as either retrieval, reasoning or decomposition error, and can be rewritten as a self-reflective trajectory that "turns" to the correct trajectory on the first incorrect step. The proposed model outperforms several baselines on in-domain and out-of-domain multi-hop search task.

**Strengths:**

1. The paper is well-written and easy to follow. The paper consists of two parts: MCTS data curation and SFT+RL training with the curated dataset, which are both very clearly conveyed. The MCTS's "exploration" and "simulation" are different from the usual use (a route from root to leaf on MCTS is not directly a solution, but only a division of tasks), but this is clearly explained in the paper and illustrated in Fig. 1.

2. The ideas are intuitive: the use of MCTS (which is also essentially a planner-executor multi-agent framework) does not only increases the possibility of successful rollouts with the limited model ability, but it also provides higher data efficiency - the "incorrect trajectories" can be recycled into trajectories with reflective behavior.

**Weaknesses:**

1. The purpose of using MCTS is to get rid of the depenence on expert large reasoning models (line 56), but the authors still use QwQ-32B, which is a much stronger model than the final MSearcher, to generate data. This design somewhat contradicts with the purpose - can the author further explain why do we not want to use expert large reasoning models in the first place, and how using QwQ-32B still supports this motivation?

2. The empirical evaluation can be improved:

a)  the ablation study does not include any experiment about the hyperparameter of MCTS, or analysis on the dynamics of data curation (e.g. how many trajectories are successful, what is the average number of steps in total, what is the average number of steps before failure for failed trajectory, what is the ratio for each type of error defined in Sec. 3.1.4, etc.)

b) In Tab. 3, the result shows that MSearcher works better with self-reflective data. However, it is unsure whether this performance difference comes from the reflective behavior, or simply because it is now trained with less data.

**Minor Weakness**

1. Fig. 1,  contry -> country.

**Questions:**

I have two questions:

1. Is the target question given in the form of multiple queries (i.e. $n>1$ in the prompt at line 90 $q=\{q_1,q_2,\dots,q_n\}$), or are all subquestions the product of MCTS?

2. in line 429, the authors mention that the performance decrease of SFT is because "a stronger bias toward tool usage introduced by the SFT data"; on the other hand, the lower average tool usage getting lower "consequently" leads to final performance decrease (line 425-426). Also, "SFT gives a higher initial tool usage, leading to better performance" (line 437-438). The effect of tool usage on final performance seems contradictory. Is more tool usage a good thing or bad thing?

---

> ### Author Response · Authors · 2025-11-17
> **Official Reply**
>
> Your suggestions are greatly appreciated. We reorganize your questions as follows:
>
> ### **Q1: Regarding the question of why use QwQ-32b.**
>
> According to results reported in the ASearcher paper, direct generation with QwQ-32B performs worse on these datasets than RL-based baselines and MSearcher.
>
> We are sorry that the statement and references in the paper were inaccurate. By “expert large reasoning models,” we primarily refer to very large and expensive models such as claude-4 or openai o1. In standard rejection sampling, generating high-quality search-augmented trajectories without such models is challenging. Using QwQ-32B would require a large number of samples.
>
> Our proposed method addresses this by decomposing complex questions, so that each sub-question can be solved by relatively low-cost open-source models (compared to claude or o1). In fact, for single-hop problems in Musique, models like Llama-3-8B-Instruct could likely suffice, but in our experiments, we used Qwen-Q-32B to ensure reliable sampling of correct trajectories.
>
> ### **Q2: Regarding the detail hyperparameter of MCTS.**
>
> We thank the reviewer for raising this point that the paper did not discuss it explicitly due to the page limit.
>
> First, it is important to note that our use of MCTS differs from standard convergence-focused MCTS. We use MCTS for data sampling, not for optimizing a score to convergence. In fact, we aim to generate diverse decompositions and rollouts rather than converge to a single high-scoring path.
> In practice, we set both the node expansion limit and rollout limit per leaf to 3 to balance efficiency, correctness, and diversity. For example, consider a 3-hop problem in Musique: a two-step decomposition creates a 3-layer tree with 9 leaf nodes. Performing 3 rollouts per leaf yields 27 rollouts in total. Typically, only one leaf yields a correct final answer; the other branches provide up to 8 diverse decomposition errors. Comparisons among rollouts of the correct leaf produce additional examples of the other two error types. Retrieval errors are rare due to the limited context (20 paragraphs) provided by Musique, so we manually insert a few rounds in the correct rollouts with slightly less relevant paragraphs. Similarly, if no rollout naturally produces reasoning errors, we insert a round with the correct context but an incorrect answer to supplement the reflective signal.
>
> Finally, each question is controlled to produce roughly 1–2 trajectories without reflection, 5–7 decomposition-reflection trajectories, 2–4 retrieval-reflection trajectories, and 2–4 reasoning-reflection trajectories, totaling around 13 trajectories per question. This corresponds to an approximate **50%** sampling efficiency (13/27).
>
> ### **Q3: Regarding the question that the performance difference in Table 3 whether comes from self-reflective signal or less data.**
>
> The current experiment indeed has a small amount of bias due to the difference in effective data size.
>
> For our constructed data, consider 4-hop questions: if one reflective step is included, the trajectory becomes 5 steps. Removing the reflective step reduces the number of single-turn dialogues in SFT by roughly 20% (since multi-turn dialogues are split into multiple single turns in SFT training). Based on our empirical observations in the experiment, a data reduction of this magnitude introduces only a very small effect on final performance—far smaller than the performance gap we observe between models trained with and without self-reflective signals.
>
> In future experiments, we can further control the total training token count, such as increase the training epoch from 5 to 6, to fully eliminate this discrepancy and provide a cleaner comparison.
>
> ### **Q4: The question expression at line 90.**
>
> This task definition section provides a general formulation for multi-hop questions. Therefore, the notation $q=\lbrace q_0,q_1, \ldots, q_n\rbrace $ should be interpreted as the ideal decomposition of any multi-hop question, rather than queries explicitly provided in the input.
>
> ### **Q5: Regarding the seemingly contradictory statement on tool usage.**
>
> Overall, we believe that increased tool usage is beneficial. In line 429, we attribute the performance drop to “a stronger bias toward tool usage introduced by the SFT data.” This refers to a mismatch between the search tools used during SFT data construction and those used during RL training: the SFT data rely on local retrieval via FAISS-based dense vector search, whereas the RL stage uses web search. In the seed dataset Musique for SFT data construction, each question is paired with a few dozen highly relevant paragraphs that serve as the retrieval corpus. This setup makes data construction easier and more efficient, but it also introduces a distributional discrepancy compared to the real web search setting.

---

> > ### Comment · Reviewer_xeVA · 2025-11-25
> >
> > Thank you for your response. If I understand correctly, the problem of Tab. 3 "excluding self-correction data" means "removing the reflection step from every trajectory", instead of "removing any trajectories that contains reflection", as the dataset includes data both with and without reflection (and the vast majority of data are with reflection according to the authors' answer to Q2). The original paper seems a bit misleading in this case. I have one follow-up question: what is the key difference between FAISS-based dense vector search vs. web search in Q5, and why would different search method causes significant distributional discrepancy?

---

> > > ### Author Response · Authors · 2025-11-27
> > > **Official Reply**
> > >
> > > You are correct about Table 3, and we appreciate you pointing this out. We will state more explicitly that the experiment removes the reflection steps within each trajectory while keeping the correct reasoning steps. This issue arises because we break each trajectory into single-turn dialogues during SFT training; thus, in the original paper, “examples” refer to these training instances.
> > >
> > > Regarding your follow-up question on retrieval discrepancy, we believe the essential difference lies in the relevance and reliability of the retrieved paragraphs. During SFT data construction, we use FAISS over the Musique-provided paragraph sets for efficiency. The questions are curated to be highly relevant to these paragraphs, so retrieval is consistently clean, accurate, and easy to use—with very high top-1 hit rates. This environment naturally encourages the model to rely on near-perfect local retrieval. In contrast, web search retrieval can be much noisier: documents may be only loosely related to the query, contain distractors, or vary in quality. This mismatch can impact model performance, especially for relatively smaller models that depend more heavily on consistent training signals.

---

### Official Review · Reviewer_b45F · 2025-10-30

**Soundness:** 3
**Presentation:** 2
**Contribution:** 2
**Rating:** 4
**Confidence:** 4

**Summary:**

This paper introduces MSEARCHER, a two-stage trained search agent designed to perform complex multi-hop question answering by combining reflective reasoning with robust tool use. The key innovation is a data construction framework based on Monte Carlo Tree Search (MCTS) that generates self-reflective reasoning trajectories for supervised fine-tuning (SFT), serving as a cold start before reinforcement learning (RL). The method leverages both correct and incorrect rollouts to train the model to recognize and correct its own errors.

**Strengths:**

1) The use of MCTS to generate diverse and self-reflective training data is creative and effective.
2) The proposed two-stage training (SFT followed by RL) addresses a common issue in RL-based agent training: instability in early stages.
3) MSEARCHER achieves state-of-the-art or competitive results on multiple datasets.

**Weaknesses:**

1) While the MCTS-based data construction is innovative, the paper lacks a formal analysis or theoretical grounding for why this approach should yield better reasoning trajectories. For instance, why is binary decomposition of sub-questions optimal? Why not allow n-ary splits or dynamic decomposition strategies? The design choices appear heuristic and would benefit from ablation studies or theoretical motivation.
2) The MCTS framework requires multiple rollouts, simulations, and tree expansions, which can be computationally expensive. The paper does not provide a detailed complexity analysis or discuss the scalability of this approach to larger datasets or more complex reasoning tasks. It is unclear how feasible this method would be for real-time applications or deployment in resource-constrained environments.
3) Although the paper evaluates on a range of QA datasets, most are still within the realm of factoid or multi-hop question answering. The evaluation lacks diversity in task types (e.g., commonsense reasoning, dialog-based reasoning, or multimodal QA). This limits the generalizability of the claims about MSEARCHER’s reasoning capabilities.
4) The agent’s performance is heavily dependent on the quality and availability of external search tools. The paper does not analyze the impact of search engine failures, biased retrieval, or noisy documents. In real-world settings, where search results may be unreliable or adversarial, the robustness of MSEARCHER is questionable and untested.

**Questions:**

1) Why is binary decomposition (splitting one question into exactly two sub-questions) enforced at every MCTS expansion node, and what evidence is provided that this restriction is optimal compared with n-ary or adaptive decomposition?
2) The reward function is relatively simple. Did you experiment with more sophisticated rewards, such as step-level correctness or a reward for synthesizing information across multiple turns?

---

> ### Author Response · Authors · 2025-11-17
> **Official Reply**
>
> Thank you for your valuable suggestions, we reorganize your concerns as follows:
>
> ### **Q1: Regarding the question of why we adopt the binary decomposition.**
>
> In principle, n-ary or dynamic decomposition is feasible, but it greatly increases system complexity and the probability of errors, potentially making it difficult to sample correct responses. In our design, we aim for simple expansion operations that can be reliably executed by general chat models without relying on specialized reasoning models—for instance, we use GLM-4-air for decomposition.
>
> If we were to use n-ary or dynamic decomposition, the model would need to determine at each step how many parts the current sub-question can be divided into, which is prone to errors and can lead to redundant or incorrect decompositions.
>
> Binary decomposition offers a simpler and more reliable alternative: the model only needs to determine whether a sub-question is single-hop, rather than estimating the quantity of n for n-ary decomposition. This ensures higher decomposition accuracy and consistent tree expansion.
>
> ### **Q2: Regarding the detailed complexity analysis of MCTS.**
> We thank the reviewer for raising this point that the paper did not discuss it explicitly due to the page limit.
>
> First, it is important to note that our use of MCTS differs from standard convergence-focused MCTS. We use MCTS for data sampling, not for optimizing a score to convergence. In fact, we aim to generate diverse decompositions and rollouts rather than converge to a single high-scoring path.
>
> In practice, we set both the node expansion limit and rollout limit per leaf to 3 to balance efficiency, correctness, and diversity. For example, consider a 3-hop problem in Musique: a two-step decomposition creates a 3-layer tree with 9 leaf nodes. Performing 3 rollouts per leaf yields 27 rollouts in total. Typically, only one leaf yields a correct final answer; the other branches provide up to 8 diverse decomposition errors. Comparisons among rollouts of the correct leaf produce additional examples of the other two error types. Retrieval errors are rare due to the limited context (20 paragraphs) provided by Musique, so we manually insert a few rounds with slightly less relevant paragraphs in the correct rollouts. Similarly, if no rollout does not naturally produce reasoning errors, we insert a round with the correct context but an incorrect answer to supplement the reflective signal.
>
> Finally, each question is controlled to produce roughly 1–2 trajectories without reflection, 5–7 decomposition-reflection trajectories, 2–4 retrieval-reflection trajectories, and 2–4 reasoning-reflection trajectories, totaling around 13 trajectories per question. This corresponds to an **approximate 50%** sampling efficiency (13/27), achieving higher sampling efficiency compared to standard rejection sampling.
>
> ### **Q3: Regarding the issue of lacking diversity in task types.**
>
> We believe that our method could be extended to other tasks, such as multimodal QA, if additional tools are incorporated. However, in the current work, we focus primarily on multi-hop and factoid question answering, which is consistent with the scope of previous studies mentioned in the paper.
>
> ### **Q4: Regarding the issue of impact of search engine.**
>
> Indeed the search module has a significant impact on QA system performance. However, search engine design is not the focus of this paper. From a controlled-experiment perspective, we use a single practical search engine for RL consistently throughout all experiments to ensure comparability.
>
> Our main focus is on enhancing reasoning from the model’s perspective. Through autonomous decision-making and reflection, the model is expected to perform multiple retrievals to improve final answers. For instance, in the case of search engine failures, the model can simply request a re-retrieval; for biased retrieval or noisy documents, the model is expected to detect and correct errors through reflective reasoning. And we further validate the contribution of reflection on retrieval with additional results on the base of ablation study 1, as demonstrated in the Q4 in response to reviewer 7uZ4.
> | Method    | HotpotQA (Acc) |
> |-----------|----------|
> | Msearcher  | 67.6 |
> |   -retri |  66.3  |
> |   -reas | 67.1 |
> |   -decom | 66.8 |
> |   -all ref | 65.7 |
> |   -sft | 63.3 |
>
> ### **Q5: Regarding the reward function.**
>
> We explored alternative reward designs, such as the relevance of retrieved content to the question and the number of useful retrievals. However, these process rewards are not always accurate or objective and can easily lead to reward hacking during training. Therefore, we primarily use final answer correctness as the reward signal.

---

> ### Author Response · Authors · 2025-11-27
> **Follow-up Message**
>
> Hello, this is a follow-up message in case you haven’t had a chance to look at our earlier response. If there are any remaining questions or points we can clarify, we’d be happy to address them. Thanks again for your time!

---

### Official Review · Reviewer_7uZ4 · 2025-11-01

**Soundness:** 2
**Presentation:** 2
**Contribution:** 2
**Rating:** 4
**Confidence:** 4

**Summary:**

MSEARCHER: Self-Reflective Search Agent Empowered by Monte Carlo Tree Search-Based Data Synthesis proposes a two-stage training framework that combines supervised fine-tuning (SFT) with reinforcement learning (RL) to improve long-horizon, multi-hop reasoning for large language models (LLMs). The key innovation lies in a Monte Carlo Tree Search (MCTS)-based data construction process that generates self-reflective reasoning trajectories, leveraging both correct and flawed rollouts to produce high-quality synthetic training data. This approach enables stable RL training, better tool-use decision-making, and strong generalization across diverse QA benchmarks. Experiments demonstrate significant improvements over state-of-the-art search agents (e.g., DeepResearcher, ASearcher), achieving 67.6% on HotpotQA and 52.0% on Frames, confirming the importance of self-reflective data for enhancing reasoning robustness.

**Strengths:**

1.  Introduces an MCTS-based framework that synthesizes self-reflective reasoning data without requiring large reasoning models, improving data diversity and efficiency.
2. Combines SFT cold-start with RL fine-tuning, effectively stabilizing early-stage training and enhancing reasoning depth.
3. Outperforms multiple advanced baselines (DeepResearcher, Search-r1, ASearcher) on both in-domain and out-of-domain multi-hop QA benchmarks.
4. Provides a clear, modular design for integrating decomposition, retrieval, and self-reflection—offering practical reproducibility and strong generalization.

**Weaknesses:**

1. Although the paper proposes a reflective data construction framework, it lacks a theoretical analysis of the convergence and sampling efficiency of MCTS in high-dimensional reasoning spaces.
2. While the paper categorizes retrieval, reasoning, and decomposition errors, it does not systematically discuss how these error types accumulate during the reinforcement learning stage.
3. Despite claiming efficiency, the paper does not provide detailed comparisons of computational resources, time costs, or scalability, leaving the practical extensibility uncertain.
4. Although partial ablation studies are conducted, the paper does not sufficiently demonstrate the independent contributions of different reflective signal types (e.g., retrieval vs. reasoning errors).

**Questions:**

See the Weaknesses.

---

> ### Author Response · Authors · 2025-11-17
> **Official Reply**
>
> We appreciate your valuable suggestions, and we reorganize your questions as follows:
>
> ### **Q1: regarding the issue of lacking analysis of convergence and sampling efficiency of MCTS.**
>
> We thank the reviewer for raising this point that the paper did not discuss it explicitly.
>
> It is important to note that our use of MCTS differs from standard convergence-focused MCTS. Here, MCTS is used for data sampling, not for optimizing a score to convergence. In fact, we aim for diverse decompositions and rollouts rather than convergence to a single high-scoring path. In practice, we observed that the model often perform too well on decomposition, producing paths very close to the gold decompositions and reducing diversity. To address this, we increase decomposition diversity by adjusting temperature and instructing the model to generate multiple alternatives in a single generation.
>
> Regarding sampling efficiency, consider a 3-hop question in Musique: if we limit each node’s expanding states (i.e., decomposition plans) to 3, a two-step decomposition creates a 3-layer tree with 9 leaf nodes. Performing 3 rollouts per leaf results in 27 rollouts total. Typically, only one leaf yields a correct final answer; the other branches provide up to 8 different decomposition errors. Comparisons among rollouts of the correct leaf produce additional examples of the other two error types. Retrieval errors are rare due to the limited context (dozens of paragraphs) provided by Musique, so we can manually insert a few rounds in the correct rollouts with slightly less relevant paragraphs. Similarly, if no rollout naturally produces reasoning errors, we insert a round with the correct context but an incorrect answer to supplement the reflective signal.
>
> Finally, we control each question to produce roughly 1-2 correct trajectory, 5–7 decomposition-reflection trajectories, 2–4 retrieval-reflection trajectories, and 2–4 reasoning-reflection trajectories, totaling around 13 trajectories per question. This corresponds to an approximate 13/27 $\approx$ **50%** sampling efficiency, balancing diversity and correctness.
>
> ### **Q2: Regarding how different error types accumulate during the RL stage.**
>
> Indeed, during training, we did not track whether each rollout contained a specific type of reflective signal, as this would introduce additional computation overhead. We plan to include such detailed statistics in future versions.
>
> Currently, we have performed a simple analysis on the trained models using the part pf the HotpotQA test set. We find that approximately 79.3% of problems contain at least one type of reflection. Among the three reflection types, the proportions are roughly 48.9% retrieval, 21.0% reasoning, and 30.1% decomposition errors.
>
> ### **Q3: Regarding the issue of computational resources, time costs, or scalability.**
>
> Regarding efficiency, we assume the reviewer is referring to the data-construction process, and thus we compare our method against standard rejection sampling.
>
> Because our access to Claude and o1 is limited, we assume both methods are conducted using QwQ-32b models for small-scale manual comparison. As discussed in Q1, the sampling efficiency of our method is approximately 50%. For standard rejection sampling, we typically use $k \approx 5$ (i.e., generate five candidates and retain one), which yields comparable data quality but rarely with self-reflection. The time cost per rollout is roughly similar between the two methods.
>
> ### **Q4: Regarding the issue of contributions of different reflective signal types.**
>
> We thank the reviewer for this suggestion. Based on the first part of our ablation study, we conducted some additional experiments.
>
> | Method    | HotpotQA (Acc) |
> |-----------|----------|
> | Msearcher  | 67.6 |
> |   -retri |  66.3  |
> |   -reas | 67.1 |
> |   -decom | 66.8 |
> |   -all ref | 65.7 |
> |   -sft | 63.3 |
>
> The results indicate that retrieval-reflection has the largest impact. This is likely because repeated retrieval allows the model to access more information, which aligns with the statistics reported in Q2—during RL, the model gradually tends to favor more useful patterns. We will further refine and expand these experimental results in the next version of the paper.

---

> > ### Comment · Reviewer_7uZ4 · 2025-11-22
> >
> > Thank you for the rebuttal, which partially addresses my concerns. However, I believe that techniques such as MCTS and DAPO have already been widely applied in search-based agent systems, and this work does not present a fundamental innovation or a paradigm-level advancement. Instead, it appears more like an engineering-oriented optimization. Moreover, the formal theoretical justification remains weak.

---

> > > ### Author Response · Authors · 2025-11-22
> > > **Official Reply to Reviewer 7uZ4**
> > >
> > > Thank you for the follow-up comments. We agree that MCTS and DAPO have been used in some prior works; however, our contribution does not lie in reintroducing these techniques, but in adapting them to the specific failure modes of LLM-based complex QA. Our work identifies a concrete limitation—LLMs’ tendency to generate short, shallow search chains—and proposes a targeted solution: a long reasoning–chain generation strategy for cold-start SFT. Its integration with an MCTS-based search strategy further provides significant gains in sampling efficiency and naturally induces self-reflective verification patterns, which we empirically show to improve factuality across diverse QA tasks. In this sense, we believe our work provides meaningful value to both practitioners and researchers working toward more reliable LLM-based QA systems.
> > >
> > > We sincerely appreciate the reviewer’s time and feedback. We hope that the clarifications above help convey the motivation, scope, and contribution more clearly, and we kindly invite the reviewer to reconsider their assessment.

---

### Official Review · Reviewer_RG1c · 2025-11-01

**Soundness:** 3
**Presentation:** 3
**Contribution:** 2
**Rating:** 4
**Confidence:** 4

**Summary:**

This paper introduces MSEARCHER, a self-reflective search agent designed to address the instability and inefficiency of training large language models (LLMs) with RL for complex reasoning tasks. The authors propose a two-stage training pipeline that begins with a supervised fine-tuning "cold start" to provide the model with a stable foundation. The core innovation is a data construction framework based on Monte Carlo Tree Search , which decomposes complex questions into smaller sub-problems. This framework generates high-quality, self-reflective reasoning trajectories by leveraging both correct and flawed rollouts from the search tree, effectively teaching the model error-correction and robust reasoning. Following the SFT stage, the agent is further trained with RL to enhance its performance. The results demonstrate that MSEARCHER significantly outperforms previous methods on multi-hop question-answering benchmarks like HotpotQA and Frames, highlighting the effectiveness of using a high-quality SFT phase to stabilize RL training.

**Strengths:**

1、The proposed method of using Monte Carlo Tree Search (MCTS) to synthesize reasoning trajectories is highly effective, with the generation of self-reflective trajectories being a particularly novel and valuable contribution.

2、This paper thoroughly explores the two-stage Supervised Fine-Tuning (SFT) and Reinforcement Learning (RL) paradigm, effectively demonstrating its power in enhancing an agent's reasoning capabilities.

3、The experiments are comprehensive and the results are significant, showing that the proposed agent consistently outperforms strong baselines on multiple challenging benchmarks.

**Weaknesses:**

1、The paper primarily quantifies the method's effectiveness through experimental results but lacks a deeper analysis, such as the underlying reasons for the observed improvements.

2、Based on the experimental results, MSearcher does not appear to have a substantial advantage, especially when compared to ASearcher.

3、Table 4 seems to indicate a performance drop of 4.8 on HotpotQA with SFT. What accounts for this decrease? Did you train a 7B-version of MSearcher, and what were its performance metrics from the base model to SFT and then to RL?

4、What would be the effect if, instead of using a complex algorithm like MCTS for data construction, a simpler method such as Rejection Sampling (RFT) were employed for SFT data generation? The authors need to elaborate on their motivation for using MCTS.

**Questions:**

Stated in Weaknesses

---

> ### Author Response · Authors · 2025-11-17
> **Official Reply**
>
> Thank you for your valuable suggestions. We reorganize your questions as follows:
>
> ### **Q1: regarding the issue of lacking a deeper analysis of underlying reasons for the observed improvements.**
>
> In addition to reporting overall performance gains, our paper also provides empirical evidence that helps explain why our method leads to these improvements, and we are happy to further clarify this analysis.
>
> Our explanation can be understood from two complementary perspectives:
>
> 1. **Training dynamics**.
> We observe a clear increase in tool-call frequency during training, mainly because the reflection component introduces additional rounds of checking for the same question. The reward curve also rises steadily, indicating that the model is learning to perform deeper and more reliable multi-step reasoning.
>
> 2. **Model behavior in generated traces**.
> In the evaluation, we also observed that the model frequently engages in reflection and multi-step verification—as shown in the case in the appendix. These qualitative differences align with the performance gains and help explain the improvement.
>
> ### **Q2: Regarding the seemingly limited improvement, especially compared to ASearcher.**
>
> We acknowledge that while our method achieves a higher overall average score, it underperforms ASearcher on a few datasets. However, it is important to emphasize that our approach focuses on improving performance through a cold-start stage using our curated SFT data, while the RL stage uses exactly the same data as DeepResearcher. In contrast, ASearcher’s gains primarily come from constructing different, high-quality RL data. Therefore, placing the two methods in the same table is not entirely fair, though it still provides a reference that both methods can improve the performance.
>
> ### **Q3: Regarding the performance drop after SFT.**
>
> As discussed in the paper, we believe the main cause is a distribution gap between the SFT data constructed from the Musique dataset and the RL training data. This effect is more pronounced in smaller models. In our early experiments with Qwen3-4B, we observed that the SFT model underperformed the instruct model on RL test sets (e.g., HotpotQA), and subsequent RL training offered little improvement compared to directly RL-trained instruct models, The and 7B model shows a similar pattern. This is likely due to overfitting in smaller models. In contrast, larger models (e.g., 14B) did not exhibit this issue, which is why we report primarily the 14B results. We appreciate the suggestion and will clarify this further in Section 5.2.
>
> ### **Q4: The motivation for using MCTS instead of rejection sampling.**
>
> In our early experiments, we found that generating a multi-step reasoning trajectory with tools by standard rejection sampling is challenging with models like Qwen-32B, and often requires a large number of samples, which is why we did not adopt a direct rejective sampling. MCTS offers several advantages:
>
> 1. **Applicability to Smaller Models**: Our approach enables smaller models to handle the binary decomposition and answering of sub-questions. Even chat models without deep thinking ability can do very well on decomposition, producing paths very close to the gold decompositions and thereby reducing diversity in the search. To address this, we deliberately increase decomposition diversity—e.g., by adjusting temperature or instructing the model to generate multiple alternatives in a single generation—to capture a wider variety of reasoning errors and reflective signals.
>
> 2. **Alignment and simplicity**: The decomposed sub-questions are preserved in topological order by the tree structure of MCTS, allowing alignment with gold sub-questions, contexts and answers without relying on additional model to score the process.
>
> 3. **Diverse reflection data**: It is acceptable if the overall rollout accuracy is not high. In fact, we welcome diverse incorrect branches, as they are valuable for generating rich reflection signals and improving the efficiency of response learning.
>
> In general, if in our method we were to retain only the correct rollouts, the effect would be equivalent to rejection sampling. However, because our approach performs tree-structured decomposition, it is applicable to smaller models and can make use of incorrect rollouts instead of rejecting them all.

---

> ### Author Response · Authors · 2025-11-27
> **Follow-up Message**
>
> Hello, this is a follow-up message in case you haven’t had a chance to look at our earlier response. If there are any remaining questions or points we can clarify, we’d be happy to address them. Thanks again for your time!

---

### Note · Authors · 2026-01-23

I have read and agree with the venue's withdrawal policy on behalf of myself and my co-authors.